# Detecting and Treating Psychosocial and Lifestyle-Related Difficulties in Chronic Disease: Development and Treatment Protocol of the E-GOAL eHealth Care Pathway

**DOI:** 10.3390/ijerph18063292

**Published:** 2021-03-23

**Authors:** Cinderella K. Cardol, Judith Tommel, Henriët van Middendorp, Yvette Ciere, Jacob K. Sont, Andrea W. M. Evers, Sandra van Dijk

**Affiliations:** 1Health, Medical and Neuropsychology Unit, Leiden University, 2333 AK Leiden, The Netherlands; j.tommel@fsw.leidenuniv.nl (J.T.); h.vanmiddendorp@fsw.leidenuniv.nl (H.v.M.); y.ciere@rdgg.nl (Y.C.); a.evers@fsw.leidenuniv.nl (A.W.M.E.); sdijk@fsw.leidenuniv.nl (S.v.D.); 2Department of Biomedical Data Sciences, Section Medical Decision Making, Leiden University Medical Center, 2333 ZA Leiden, The Netherlands; j.k.sont@lumc.nl; 3Healthy Society, Medical Delta, 2629 JH Delft, The Netherlands

**Keywords:** lifestyle adherence, psychosocial adjustment, chronic disease management, intervention development, eHealth, screening, web-based cognitive-behavioral therapy, tailored personalized treatment, Behavior Change Wheel, co-creation

## Abstract

Many patients with lifestyle-related chronic diseases find it difficult to adhere to a healthy and active lifestyle, often due to psychosocial difficulties. The aim of the current study was to develop an eHealth care pathway aimed at detecting and treating psychosocial and lifestyle-related difficulties that fits the needs and preferences of individual patients across various lifestyle-related chronic diseases. Each intervention component was developed by (1) developing initial versions based on scientific evidence and/or the Behavior Change Wheel; (2) co-creation: acquiring feedback from patients and health professionals; and (3) refining to address users’ needs. In the final eHealth care pathway, patients complete brief online screening questionnaires to detect psychosocial and lifestyle-related difficulties, i.e., increased-risk profiles. Scores are visualized in personal profile charts. Patients with increased-risk profiles receive complementary questionnaires to tailor a 3-month guided web-based cognitive behavioral therapy intervention to their priorities and goals. Progress is assessed with the screening tool. This systematic development process with a theory-based framework and co-creation methods resulted in a personalized eHealth care pathway that aids patients to overcome psychosocial barriers and adopt a healthy lifestyle. Prior to implementation in healthcare, randomized controlled trials will be conducted to evaluate its cost-effectiveness and effectiveness on psychosocial, lifestyle, and health-related outcomes.

## 1. Introduction

Lifestyle-related chronic diseases—such as type 2 diabetes, cancers, and cardiovascular, kidney, and chronic respiratory diseases [1,2]—form the leading causes of death, accounting for 71% of global mortality in 2016 [2]. Although these diseases largely differ regarding treatment regimens, disease-specific guidelines have one thing in common: They stress that this mortality could be greatly lowered if patients with such lifestyle-related diseases would adhere to a set of key healthy lifestyle behaviors [3], including engaging in regular physical activity, keeping a healthy weight and diet, refraining from smoking, and adhering to medication prescriptions (e.g., [1,4]). Engaging in these behaviors could also diminish cardiovascular complications, hospitalizations, comorbidities, and physical as well as psychological disease burden [3]. However, sustained adherence to healthy lifestyle behaviors is only achieved by a minority of patients: In multiple cohort studies, less than 5% of patients reached all lifestyle guidelines, even including individuals who already had experienced a coronary heart disease or stroke event [5,6].

These findings show that it is very difficult to adjust to chronic disease and adopt an active, healthy lifestyle. It requires challenging coping skills, such as accepting diagnosis and prognosis, managing physical and social implications, and changing long-standing habits. These challenges influence patients’ psychosocial functioning: Many experience psychological distress, that is, symptoms of depression or anxiety, including sadness, loss of interest, irritability, nervousness, or restlessness [7,8,9]. Psychological distress symptoms often go hand in hand with deficits in energy, self-regulatory resources, memory, motivation, optimism, self-efficacy, and social support. These problems in patients’ psychosocial functioning may form strong barriers for engagement in a healthy lifestyle [10,11,12]. For instance, recent systematic reviews showed that depressive symptoms among patients with type 2 diabetes were negatively associated with physical activity and dietary adherence [13], and that both depressive and anxiety symptoms predicted medication non-adherence among kidney transplant recipients [14]. Additionally, many pulmonary and cardiac patients seem to avoid physical activity due to fears about physical symptoms, such as not being able to breathe or having a cardiac event [15]. Thus, adequate psychosocial adjustment to chronic disease may be a prerequisite for lifestyle adherence.

Even though psychological distress may hamper the uptake of a healthy lifestyle, most existing support strategies focus either solely on diminishing psychological distress or only on improving lifestyle behaviors. On the one hand, mental healthcare mainly focuses on treating psychological distress symptoms, but their interaction with chronic somatic disease and its lifestyle management is not always sufficiently taken into account; on the other hand, lifestyle interventions in the medical setting tend to pay insufficient attention to psychological burden that may hinder the engagement in a healthy lifestyle [16,17]. It would be valuable to integrate support strategies for both psychological distress and lifestyle in chronic disease: This could not only diminish psychosocial barriers to improve adherence to healthy lifestyle behaviors, but vice versa, the uptake of healthy and active lifestyle behaviors could also help to reduce psychological distress [10]. For instance, enhancing mood may establish the energy and motivation needed to engage in physical activity, and in turn, activity may alleviate anxiety and depressive symptoms [18]. Literature suggests that integrated treatments, aimed at bi-directional improvements in psychosocial functioning and lifestyle management, could be more effective than one-sided interventions in improving physical as well as psychological outcomes and patients’ quality of life [10,13,17].

A first step to support patients with both psychosocial and lifestyle adjustments is to identify patients who experience difficulties in these areas, and are therefore at increased risk for poor mental and physical health outcomes, such as low positive affect, quality of life, and disease progression or complications [19,20]. However, the literature shows that, in busy clinical practice, it is challenging for medical health professionals to assess and discuss their patients’ psychosocial difficulties and cues that may indicate non-adherence, which may not be readily observable [21]. Patients and providers may normalize psychosocial difficulties as a “logical” consequence of chronic disease, or even attribute psychological symptoms to physical health conditions [22]. Both could lead to under-diagnosis and under-treatment of psychological health problems. For instance, in a recent study among patients with cardiopulmonary conditions, it was found that only 32% of patients who met diagnostic criteria for depression and 9% who met criteria for anxiety actually had those diagnoses documented in their electronic medical records [23]. A screening tool could aid professionals to identify psychosocial problems—as well as lifestyle-related difficulties—that may be overlooked otherwise. Such a tool could also facilitate addressing those difficulties in consults and selecting patients who may benefit from specialized support strategies [1,19]. Recent studies suggest that online completion of questionnaires can make such a screening process more complete and efficient compared with paper-and-pencil completion [24]. Furthermore, patients evaluated online screening positively, for example, because it could contain direct visual feedback that provides them insights into their own health [24,25].

An online modality may not only be a useful option for a screening tool, but also for specialized support strategies: When supporting patients with lifestyle-related chronic diseases in psychosocial and lifestyle adjustments, potential barriers for face-to-face support are, for instance, physical limitations that complicate traveling to therapy or perceived stigma related to mental support seeking [16]. To overcome such barriers and improve accessibility and acceptability of support, electronic health (eHealth) interventions, and specifically Internet-delivered cognitive-behavioral therapy (iCBT), may be a solution [19]. Additional advantages of eHealth and iCBT treatments are enhanced flexibility and tailoring to personal preferences, the accessibility of therapy from the privacy and comfort of one’s home, and a relatively easy application of learned techniques in patients’ own environments [16,26]. A recent systematic review showed a high feasibility of and satisfaction with eHealth interventions among patients with chronic kidney disease [27]. Furthermore, promising effects of iCBT have been shown by a growing body of evidence among patients with chronic diseases on physical and psychological outcomes as well as health-related quality of life, especially when interventions are guided by a therapist: Systematic reviews found moderate effects of therapist-guided iCBT on depression and anxiety, with effect sizes comparable to face-to-face CBT [28,29]. The largest effects have been found for interventions that are tailored to patients’ individual complaints and needs [29].

In conclusion, guided and tailored eHealth care pathways, that is, complex interventions that combine screening and integrated psychosocial and lifestyle support strategies, could aid patients with lifestyle-related chronic diseases. Therefore, the overall objective of this study was to develop such an eHealth care pathway, including (1) a screening tool with questionnaires to identify patients who experience psychosocial and lifestyle-related difficulties and to tailor the intervention, as well as personal profile charts to visualize screening outcomes, and (2) lifestyle treatment modules embedded within existing guided and tailored iCBT to treat psychological distress, diminish psychosocial barriers, and promote psychosocial facilitators for engagement in healthy and active lifestyle behaviors. In this paper, the systematic development per intervention component is described, as well as the final version of the tailored eHealth care pathway for application in patients with chronic kidney diseases.

## 2. Development

The eHealth care pathway was systematically developed by a research team of health psychologists working in academia and therapy practice (C.K.C., J.T., H.v.M., Y.C., A.W.M.E., and S.v.D.), as well as a clinical epidemiologist/medical decision-making scientist (J.K.S.), based on previous experiences in the development of (eHealth) interventions for patients with chronic diseases [30,31]). The eHealth care pathway was developed for different research projects among patients with lifestyle-related chronic diseases, including patients with chronic kidney disease (CKD; E-GOAL study. Netherlands Trial Registry, study number NL7338, medical ethics committee METC-LDD reference numbers P17.090 and P17.172), end-stage kidney disease (ESKD; E-HELD study. Netherlands Trial Registry, study number NL7160, METC-LDD reference number P18.013), and patients with lung, stomach, intestine, and liver diseases [16]. Some characteristics of the eHealth care pathway, such as specific questionnaires, may differ across research projects depending on the patient population. Here, the development of the version used in the E-GOAL study is presented. This study’s version of the eHealth care pathway is developed for an effectiveness evaluation in a randomized controlled trial, conducted at the nephrology departments of four medical centers in the Netherlands.

For each component of the intervention, the development was conducted by: (1) Using scientific evidence and expertise from our research team to develop initial versions of the intervention components; (2) acquiring feedback from users (i.e., patients with lifestyle-related chronic diseases and health professionals) regarding usability and feasibility; and (3) revising and refining the intervention components. The second and third stage were conducted in several iterations if needed, to fully address users’ needs and preferences. See Figure 1 for an overview.

### 2.1. Screening Tool

The screening tool consists of (1) screening questionnaires to identify patients with an increased-risk profile—who experience psychosocial and lifestyle-related difficulties—as well as questionnaires to tailor the intervention, and (2) personal profile charts to visualize screening results. The screening tool was embedded in the online platform “PatientCoach”, an eHealth application to support patients with chronic somatic diseases, developed and hosted at Leiden University Medical Center (LUMC) [32].

### 2.2. Screening Questionnaires: Increased-Risk Profile Identification and Intervention Tailoring

#### 2.2.1. Development Screening Questionnaires Stage 1: Developing Initial Version

In order to limit the burden of filling out questionnaires, we decided to use a stepped approach in which the screening questionnaires are divided into two successive parts.

##### Part 1: Questionnaires for Increased-Risk Profile Identification

The first, brief part is used to select patients with an increased-risk profile, that is, patients who experience psychosocial and lifestyle-related difficulties, who are thus at increased risk of poor health outcomes. These patients are most likely to benefit from the iCBT treatment targeting psychosocial determinants of healthy lifestyle behaviors [19]. To screen for psychosocial difficulties that potentially form barriers for healthy lifestyle behaviors, we decided to measure depressive symptoms, anxiety symptoms, fatigue, and health-related quality of life. These domains have been prioritized for improvement and as barriers for lifestyle adherence by patients with lifestyle-related chronic diseases [12,33]. Furthermore, this set of psychological, social, and physical domains provides patients and health professionals a summarized overview of a patient’s mental and physical health status [19]. To screen for lifestyle-related difficulties, we included physical activity, body mass index (BMI), eating behaviors, smoking, and medication adherence. We selected questionnaires to measure these psychosocial and lifestyle variables, based on their validity and reliability in populations with chronic diseases, and their feasibility for users (i.e., low response burden and good comprehensibility). Detailed information regarding the selection of questionnaires can be found in Appendix A.

Based on thorough discussion and previous experience of our research team, it was decided that patients were eligible for the iCBT treatment if they showed (1) at least mild psychological distress scores and (2) at least 1 suboptimal (i.e., unhealthy) lifestyle behavior or lifestyle-related outcome (i.e., BMI). To identify psychosocial difficulties, we used the original cut-off points of the psychological distress questionnaires to indicate at least mild depressive or anxiety symptoms. We based the cut-off points for suboptimal lifestyle behaviors on international recommendations for populations with lifestyle-related chronic diseases (e.g., <150 weekly minutes of physical activity, BMI ≥ 25). More detailed information regarding the cut-off points used can be found in Appendix A.

##### Part 2: Questionnaires for Intervention-Tailoring

The second, complementary part of the screening questionnaires has to be filled out only by patients with an increased-risk profile who are eligible for the iCBT treatment, to gather more in-depth information for tailoring the iCBT treatment to their needs and priorities [19]. We included scales regarding different areas of behavioral, psychological, social, and physical functioning. Furthermore, we developed a short Personalized Priority and Progress Questionnaire (PPPQ) to measure patients’ personal priorities for improvement as well as actual subjective improvements over time in different areas of functioning (7 items; e.g., “During the past 2 weeks, to what extent did you experience limitations regarding tiredness or sleeping problems?”) and lifestyle behaviors (5 items; e.g., “During the past 2 weeks, to what extent did you manage to eat healthily?”). This questionnaire is based on validated goal setting instruments [34,35,36]. More information about the included scales can be found in Appendix A.

#### 2.2.2. Development Screening Questionnaires Stage 2: Acquiring User Feedback

Cognitive interviews took place to evaluate the comprehensibility of the PPPQ, and to determine whether the questions are consistently interpreted as intended among different patient groups [37]. We purposively recruited patients at the Department of Nephrology in a Dutch hospital in collaboration with health professionals. Eight individuals (5 male) of 18 years of age or older with a diagnosis of CKD (*n* = 4) or ESKD (*n* = 4) were invited to participate in a 30-minute session where they completed a paper-and-pencil questionnaire and were cognitively interviewed about the items. We held two rounds of cognitive interviews. In the first round, 4 patients were interviewed. The interview moderator (C.K.C. or J.T.) read each item out loud with the possibility for the participant to read along. Participants were first invited to think aloud to encourage an open-ended dialogue. After each response, the interview moderator used general and item-specific verbal probes to address specific items and issues regarding interpretation (e.g., “Without looking at the question again, could you explain in your own words what was asked?”) and comprehension (e.g., “What does the term X mean to you?”, “Did the question contain any difficult words?”) of instructions, items, and response options. Additionally, the interviewer took notes and answered questions based on observation of the respondent (e.g., “Did the respondent seem to have any difficulty using the response options?”).

The interview moderators documented a summary of each cognitive interview in a spreadsheet. This file contained difficulties in comprehension and interpretation (e.g., misunderstanding or uncertainty in the meaning of items), observations, and participants’ suggestions for changes in difficult-to-understand items. After an interview round, the interview moderators discussed the problems encountered and how they could be corrected, and H.v.M. and S.v.D. reviewed the proposed modifications before the questionnaires were adapted. After this refinement, we repeated stages 2 and 3 of development, that is, C.K.C. and J.T. tested the adaptations in subsequent interviews with 4 other participants and repeated the analysis procedure.

#### 2.2.3. Development Screening Questionnaires Stage 3: Revising and Refining

Regarding the PPPQ, most items functioned as intended. In the items assessing priorities for functioning, 2 out of 7 items were revised to increase clarity and consistency of interpretation. An item about “fatigue and/or sleeping problems” caused confusion for a participant who did experience fatigue but did not suffer from sleeping problems. Therefore, “and/or” was replaced by “or”. The item “To what degree do you experience limitations in your social environment (e.g., in communication with others or dependence on others)” was found too broad to answer and was inconsistently interpreted. To clarify its meaning, the examples were specified in more detail (“e.g., communication about your needs and wishes, asking or receiving support, or dependence on others”). Last, for some participants, it was unclear for which disease or condition they should answer the questions about their experienced limitations. For instance, for the item “To what degree do you experience limitations in the area of pain?”, one participant was unsure whether to focus only on pain due to kidney disease, or also on pain due to an eye operation. To avoid this issue, “due to your [kidney] disease” (replaceable with other lifestyle-related diseases) was added to the instruction text. None of the items assessing priorities for lifestyle required revisions.

In the second interview round, no new issues with instructions, items, or response options were detected. The final version and validation of the PPPQ will be described in more detail in another manuscript by the research team (J.T., C.K.C., S.v.D., A.W.M.E., and H.v.M.), which is currently in preparation.

### 2.3. Personal Profile Charts to Visualize Screening Results

#### 2.3.1. Development Personal Profile Charts Stage 1: Developing Initial Version

Our research team agreed that two types of charts would be needed: A chart showing an overview of scores in each domain of functioning and lifestyle to visualize a patient’s current status (hereafter indicated as a profile chart), as well as a chart of measurements at different time points to monitor progress over time per domain (hereafter indicated as a monitor chart). We evaluated several prototypes for the profile chart and the monitor chart. For the profile chart, we selected two existing charts (a visual representation of scores in a wheel and in balloons) that were developed and investigated for other patient populations, respectively, within our research team (in collaboration with Netherlands Organisation for Applied Scientific Research) and by other researchers [38], and we designed one chart (a visual representation of scores in thermometers) in collaboration with health professionals within our research team. To visualize the monitor charts, we designed a line chart and a bar chart. Appendix A contains the prototypes.

#### 2.3.2. Development Personal Profile Charts Stage 2: Acquiring User Feedback

We conducted semi-structured interviews to evaluate the feasibility of each chart. Purposive recruitment took place of patients at the Departments of Gastroenterology (*n* = 2), Pulmonology (*n* = 3), and Nephrology (*n* = 7), in collaboration with health professionals. Nine of the 12 patients were male and their ages ranged from 40 to 82 years. Additionally, two nurse practitioners (both female) from the Department of Nephrology were interviewed. Participants had different levels of experience with online tools and patient portals. We held two rounds of feasibility interviews, with a total of 14 participants. In the first round, 10 patients and the two nurse practitioners were invited to participate in a feasibility interview with a duration of 15 to 30 min, in which the interview moderator (C.K.C. or Y.C.) showed respondents the charts on paper one by one. With each chart, the moderator asked questions about comprehension and interpretation (e.g., “What do you see?”). Then, they asked the participants to write down plus and minus symbols on the different parts of the chart to indicate their positive and negative impressions. Afterwards, the participants were invited to verbally elaborate on the pluses and minuses and the interviewer asked questions about feasibility (e.g., “What do you think of the design?”, “Does the information in the chart fit your needs?”, “What would you do differently?”). Last, participants were invited to choose their preferred design.

The interview moderators documented a summary of each feasibility interview in a spreadsheet. This file contained the first impression, positive remarks, improvement areas, suggested modifications, and preferred designs expressed by each participant. Subsequently, the researchers discussed the outcomes, selected the profile and monitor chart that received most votes, and adapted the designs by incorporating the respondents’ feedback. Since some major changes were made, we established another iteration, i.e., stages 2 and 3 of development were repeated: A second interview round took place among two patient members of the E-GOAL study group, one male and one female, both from the Department of Nephrology. In addition to the questions about comprehensibility and feasibility, they were asked what they would find the best way of showing the charts to users (e.g., online or on paper, with a health professional present or not). Afterwards, final refinements were made.

#### 2.3.3. Development Personal Profile Charts Stage 3: Revising and Refining

In general, participants were rather positive about the use of personal profile charts as a tool in patient–provider communication, to gain insights into patient health and areas that need attention, and to set goals and action plans for improvement. For the profile chart, 9 participants preferred the thermometers over the wheel (2 votes) and balloons (1 vote). For the monitor chart, 10 participants preferred the line charts over the bar charts (1 vote; 1 participant did not have a specific preference). The designs of the thermometers and line charts were found clearest and most suitable for a hospital setting. The research group selected the profile and monitor charts that received most votes (see Figure 2).

Even though there was quite some consensus between participants about the preferred charts, they also provided feedback and suggestions for improvement. First, domain definitions were added to the profile chart, shown when users would position their mouse cursor on the domain. Regarding the profile chart, participants suggested a horizontal positioning of bars (instead of thermometers) and domain names, to diminish confusion and improve readability. Last, for both the profile and monitor chart, two participants found the different color tones, gradually changing from red to green, unclear. It was preferred to use three traffic light colors, which are easier to distinguish.

In the second interview round, only minor problems were detected and a final refinement took place. Both participants stated that it would be useful for patients to see their questionnaire results in personal profile charts directly after filling in the questionnaires, that is, without a health professional present, provided that there would be a possibility to contact a professional in case of any questions about the results. Furthermore, they stated that the personal profile charts should be presented both online and on paper, for people who find it difficult to navigate in online patient portals. This feedback was incorporated in the final eHealth care pathway. After developing the content of the screening tool (questionnaires with cut-off points for increased-risk profiles, as well as personal profile charts), it was built into the eHealth application PatientCoach [32], as introduced before. The tool was extensively tested before patients were invited for usage.

### 2.4. iCBT Treatment

For patients who were identified by the screening tool to have an increased-risk profile, and thus eligible for the iCBT treatment, our research team developed lifestyle self-management modules. These lifestyle modules were embedded within the existing generic guided and tailored iCBT intervention “E-coach”, which already contained modules to treat psychosocial difficulties related to chronic somatic disease. E-coach was developed by the research group of Prof. A. W. M. Evers (A.W.M.E.) at Leiden University and Radboud university medical center, based on evidence-based face-to-face CBT for patients with chronic somatic conditions [30,31]. The effectiveness of this iCBT was demonstrated in randomized controlled trials in different patient populations [30,31].

### 2.5. Treatment: Lifestyle Modules

#### 2.5.1. Development Treatment Stage 1: Developing Initial Version

To develop the initial version of the lifestyle modules, we used the Behavior Change Wheel (BCW) guide [39]. The BCW is a framework for designing interventions, which integrates 19 existing behavior change theories. It consists of eight steps to guide intervention design [39]. We broadly followed these steps (see Figure 3).

In steps 1 to 3 of the BCW, researchers usually identify the specific behavior that needs to change by (1) defining the problem in behavioral terms, (2) selecting, and (3) specifying the target behavior by answering the following questions: What behavior needs to change, who needs to perform it, what do they need to do differently, when and where do they need to do it, how often, and with whom? As described before, the answers to most of these questions are quite well established in international guidelines and existing literature from various lifestyle-related chronic diseases (e.g., [1,4]). We also took the likelihood of behavior change within an intervention into account (i.e., by exploring whether previous intervention studies have been successful in bringing about the desired lifestyle changes). Table 1 summarizes the target behaviors. Further specification of the target behaviors for an individual patient depends on the person and disease characteristics (e.g., physical activity should be compatible with a patient’s health and tolerance). Thus, within the intervention, the target behavior should be further tailored to individual needs.

In step 4, we conducted eight focus groups among patients with non-dialysis-dependent chronic kidney disease (*n* = 24) and their health professionals (*n* = 23) to gain a deeper understanding of factors that may influence the target lifestyle behaviors. Barriers and facilitators for engagement in healthy lifestyle behaviors were explored, as well as intervention strategies needed to address those. Three researchers (C.K.C., S.v.D., and a physician researcher in nephrology) analyzed transcripts using thematic analysis. The codes from the inductive analysis were deductively mapped onto the domains of the Theoretical Domains Framework (TDF, e.g., knowledge, intentions, emotion) and structured onto three overarching components that are considered essential for behavior and behavior change to occur: Capability, Opportunity, and Motivation [the ‘COM-B system’; 39]). Table 2 includes an overview of the main barriers and facilitators for healthy lifestyle behaviors found in the focus group study, which has been reported in detail elsewhere [12].

In step 5, C.K.C. listed all potentially relevant intervention functions, that is, methods by which an intervention may change behavior (e.g., education, training, environmental restructuring), by linking the TDF domains identified in step 4 to the intervention functions that are most likely to affect behavior change for each domain, as described in the BCW guide [39]. Then, C.K.C. and S.v.D. evaluated the relevant intervention functions using the affordability, practicability, effectiveness and cost-effectiveness, acceptability, side effects/safety, and equity (APEASE) criteria, to select the most appropriate intervention functions. Education (increasing knowledge and understanding to enhance patients’ capability to change behavior), enablement (increasing means and reducing barriers to enhance patients’ opportunity and motivation to change behavior), persuasion (inducing positive or negative feelings and stimulating action to enhance patients’ motivation to change behavior), and incentivization (creating expectation or reward to enhance patients’ motivation to change behavior) were selected as the intervention functions most relevant for the lifestyle modules in the iCBT treatment (see also Table 2). The remaining intervention functions were regarded as unfeasible to implement within a web-based intervention targeted at individual patients.

In step 6, as described in the BCW guide, for each intervention function identified in step 5, policy categories (e.g., service provision, guidelines, marketing) should be selected that are likely to be appropriate in supporting the intervention functions. However, since the lifestyle modules were intended to be built into the existing iCBT intervention E-coach [30,31], the policy category service provision was predefined. Service provision is an adequate policy category to carry out the selected intervention functions [39].

In step 7, the BCW guide describes a taxonomy of 93 behavior change techniques (BCTs, e.g., goal setting, social support, reframing), the “active ingredients” of behavior change. The BCW guide provides a list of potentially adequate BCTs (version 1, also indicated in the literature as BCTT v1) for every intervention function. Given the relevant intervention functions selected in step 5, C.K.C. created an initial longlist of potential BCTs per TDF domain. Then, C.K.C. and S.v.D. shortlisted the most appropriate BCTs, based on an evaluation against the APEASE criteria, the most commonly used and investigated BCTs that are likely to bring about the desired behavior changes, and the previous experiences of our research team [39,40,41]. Afterwards, in order to make the intervention more effective and tailored to a patient’s lifestyle behavior change process, the BCTs were organized among 3 modules, representing different stages of behavior change, in accordance with stage theories [42]: Module 1: “Goals Exploration” (stages of contemplation and decision), Module 2: “Goals in Action” (stages of planning and action), and Module 3: “Goals Persistence” (stages of evaluation and maintenance). The selected BCTs and structure among the modules can be found in Table 2.

In the final step 8, the BCW guide recommends researchers to consider different modes of delivery for the intervention (e.g., face-to-face vs. web-based; individually vs. group). Since the lifestyle modules were embedded within the existing iCBT intervention E-coach [30,31], the researchers only partly had to engage in this step. In E-coach, patients with chronic somatic diseases complete an online trajectory of one or more treatment modules (e.g., about mood, social functioning, or physical complaints) at home, and receive regular feedback from their therapist via text messages or by telephone. Treatments using E-coach start with a face-to-face intake session, can be online or blended (with additional face-to-face sessions), and are tailored to patients’ personally relevant goals. For the newly developed lifestyle modules, we decided to employ the same online delivery mode, guided by a trained health psychologist. Experiences from our research team were used to decide on the duration of the intervention (i.e., 3–4 months) and on the inclusion of a possibility to offer additional sessions by telephone or face-to-face, in case a therapist would consider this beneficial for a patient.

#### 2.5.2. Development Treatment Stage 2: Acquiring User Feedback

An initial version of the lifestyle modules was developed and its feasibility was tested among a patient with CKD (kidney–pancreas transplant recipient, male), a patient with osteoporosis and cured breast cancer (female), and a healthy control (female). The participants were invited to set a personally relevant goal related to their lifestyle, and to work through a paper-and-pencil workbook of the modules within 1–2 weeks (without guidance). They were asked to write down any feedback on comprehensibility, usability, and acceptability of each component (e.g., psychoeducational text or exercise) in the workbook. After finishing the modules, participants filled out a questionnaire with a few open questions per module, including “Which component did you find most/least useful and appealing?” and “What would you definitely change?” Finally, the first author (C.K.C.) conducted a 15–30 min interview with each participant to further discuss their experiences and ideas. Feedback was summarized and discussed within our research team (C.K.C., S.v.D., and A.W.M.E.) and adaptations were made.

#### 2.5.3. Development Treatment Stage 3: Revising and Refining

In general, all three participants indicated that the modules were easy to comprehend, and written in a clear and positive language. Two participants stated that the option to get support or feedback, specifically when setting goals and creating action plans, is vital. They also positively evaluated the layout and structure of the modules and exercises: “It is well structured! A logical sequence of theory and exercises, and steps in the process [of behavior change] in which they [patients] will engage.” Regarding the content of the modules, two participants stressed the importance of the motivation-enhancing BCTs in the first module. They also found the exercise about strengths (BCT on valued self-identity) in the third module very appealing and valuable. Additionally, one participant found the examples and practical advices very motivating and feasible to put into practice. The same participant positively evaluated the diaries throughout the modules: “Good tools, clear and easy to use.”

All participants had some minor suggestions for improvement. Two participants reflected on the repetition of goal setting BCTs: “Goal setting appears in two exercises, with different explications and examples. This may be confusing.” Therefore, the two exercises were merged. Furthermore, with regard to the knowledge-enhancing BCTs, one participant suggested to refer to reliable sources (e.g., from the government) for additional practical and factual information about healthy lifestyle behaviors, and to encourage patients to consult a specialist (e.g., physiotherapist or dietician) for specific, personalized information on feasible lifestyle adaptations. Since these lifestyle modules are embedded in an iCBT intervention and thus mainly focused on behavior change from a psychosocial perspective, this suggestion was followed.

Since no major adaptations needed to be made, stages 2 and 3 of development were not repeated. That is, after refining the lifestyle modules based on the user feedback, the modules were built into the eHealth application E-coach [30,31], as introduced before.

## 3. Results

The final version of the tailored eHealth care pathway for patients with lifestyle-related chronic diseases is depicted in Figure 4. First, patients receive an invitation by email with a link to a personal “to do list” in the eHealth application PatientCoach, where they can complete the screening questionnaires. All patients fill in the brief screening questionnaires, to identify whether they have an increased-risk profile, i.e., whether they are at increased risk of poor health outcomes due to psychosocial and lifestyle-related difficulties. All patients can review the results of the brief screening in their personal profile charts in PatientCoach (see Figure 5 for an example), and receive a paper version by mail. 

The system automatically detects increased-risk profiles, by identifying patients who experience at least mild psychological distress and at least one suboptimal lifestyle behavior. For these patients, the complementary questionnaires—assessing specific areas of patients’ behavioral, psychological, social, and physical functioning to tailor the intervention to personal needs and priorities—appear in their to do list directly after completing the brief screening questionnaires.

Patients who show an increased-risk profile are invited by mail and telephone to receive tailored and guided blended CBT treatment using the eHealth application E-coach. This treatment starts with a face-to-face intake session of an individual patient with a therapist, that is, a trained health psychologist, which can take place in the patient’s medical center. This initial session includes an assessment of a patient’s physical, psychological, and social functioning and their interactions, guided by the personal profile charts and complementary screening results [19]. That is, by using clinical reasoning to combine and interpret the screening and intervention-tailoring questionnaires, the therapist obtains insights into the magnitude of psychosocial and lifestyle-related adjustment problems, relationships between co-occurring problems and symptoms, and their context (e.g., psychological aspects, personality characteristics, and social support). Combined, these insights indicate treatment priorities, a patient’s vulnerabilities (e.g., neuroticism or pessimism) and resilience factors (e.g., high self-efficacy or motivation) to address in treatment [19]. With that information, the therapist and patient discuss which psychosocial difficulties form barriers for which lifestyle behaviors, explore a patient’s resources that may facilitate change (e.g., based on questionnaires regarding personality characteristics and social support), and determine a patient’s priorities for improvement (e.g., based on the PPPQ). With this information, the therapist aids the patient in formulating two to three personally relevant psychosocial and lifestyle goals, and introduces the eHealth application E-coach. Thereafter, during the next 3 to 4 months, patients in treatment systematically go through several treatment modules (e.g., regarding mood, social environment, fatigue, or lifestyle; see Figure 6 for an example) matching their personal goals. Modules include psychoeducational texts and exercises based on cognitive–behavioral BCTs. Patients work through the modules at home and receive regular (e.g., weekly or bi-weekly) personalized feedback from their therapist via a secured message box within E-coach. If needed, the treatment can be complemented with telephone or face-to-face appointments.

After completing the personalized modules, patients go through a final module about relapse prevention and long-term goals, to promote maintenance of the behavior changes after treatment. In this module, patients also write a letter to themselves regarding their achievements. Afterwards, they have a final telephone appointment with their therapist to evaluate the trajectory. The exact duration of a trajectory is tailored to the number of treatment goals and the adequate pace for the individual patient.

After finishing the treatment, patients complete the screening questionnaires again and receive profile and monitor charts (see Figure 7 for an example) to see treatment effects and progress. At follow-up (e.g., 3 months after finishing the treatment), this screening questionnaire completion is repeated and patients receive an email from their therapist including their own letter to themselves, as a reminder and booster to maintain their new healthy habits.

## 4. Discussion

In the present paper, we described the systematic development of a generic eHealth care pathway, tailored to the needs of patients with lifestyle-related chronic diseases. The eHealth care pathway facilitates both psychosocial and lifestyle adjustments, which are important to reduce disease burden and risks of adverse health outcomes [3,10]. The eHealth care pathway comprises (a) a screening tool with questionnaires to identify patients who experience psychosocial and lifestyle-related difficulties and personal profile charts to visualize screening outcomes, as well as (b) tailored and guided lifestyle self-management modules alongside iCBT to treat psychological distress, diminish psychosocial barriers, and promote psychosocial facilitators for engaging in an active and healthy lifestyle. Each component was developed in three iterative stages of creating initial versions, acquiring user feedback, and further refinement. The creation of the initial versions was guided by scientific evidence and the BCW framework for intervention development. To acquire feedback from users (i.e., patients and health professionals), cognitive interviews, feasibility interviews, and focus groups took place.

In order to develop an eHealth care pathway that fits the priorities and preferences of its end users, we undertook a systematic and user-centered approach. Below, we elaborate on several characteristics of the intervention development, that is, on the advantages of using a theory-based framework and co-creation methods. First, although evidence for an association between theory use and increased intervention effectiveness compared to non-theory-based interventions is currently inconsistent, using theory-based frameworks is being promoted, since it is certainly beneficial to guide intervention design, evaluation, and optimization [43]. In the development of our eHealth care pathway, following the pre-determined steps of the BCW made it possible to systematically consider a wide range of options and BCTs for the intervention, to incorporate the ones that meet the needs of patients with lifestyle-related chronic diseases [39]. Second, multiple systematic reviews suggested that early involvement of patients, professionals, and other stakeholders in development processes is a prerequisite for successful and sustainable implementation of eHealth interventions within a medical organization [27,44,45]. We employed several co-creation methods that involved patients with different lifestyle-related chronic diseases (including kidney, lung, stomach, intestine, and liver diseases) and their health professionals, in order to develop an eHealth care pathway that is suitable for a broad range of potential end users.

A potential strength of the eHealth care pathway is its flexibility for usage in patient populations with different lifestyle-related chronic diseases [19]. Trial results indicate the feasibility and effectiveness of other versions of the screening tool and the iCBT treatment among individuals with asthma, psoriasis, and rheumatoid arthritis [30,31,32]. Furthermore, the iCBT intervention is already being applied in clinical practice, as part of regular CBT for individuals with a broad range of chronic diseases in the Netherlands (reimbursed by insurance companies), which is also a promising sign for the generalizability of the E-GOAL eHealth care pathway. Generic or transdiagnostic interventions that are applicable across various chronic diseases are becoming more relevant since multimorbidity (i.e., the co-occurrence of two or more chronic diseases in the same person) is an increasingly prevalent concern. This often results in challenges with regard to adequately tailored patient-centered care, for instance, due to fragmentation of healthcare provision [46]. A generic approach such as our eHealth care pathway goes beyond diagnoses and disease-specific support, and is therefore adequate for patients with different or multiple lifestyle-related chronic diseases. To assure that disease-specific concerns are taken into account, screening and treatment can be tailored by addressing specific symptoms (e.g., a module about itch may be relevant for a patient with ESKD, but may be left out for someone with CKD or lung complaints). Furthermore, as unhealthy lifestyle behaviors are interrelated and often occur together, the multifactorial approach in which multiple behaviors can be addressed at once could result in a greater reduction of health risks than a focus on a single lifestyle issue [47,48]. An additional advantage of our intervention is that it addresses (not necessarily disease-specific) psychosocial and lifestyle-related difficulties simultaneously. Recently, it has been recommended to implement treatments that synergistically target mental health needs and disease self-management of patients with chronic diseases [49], and thus not only take into account physical, but also mental comorbidities. Given these recommendations, the eHealth care pathway may be a valuable innovation.

The eHealth care pathway has not only been tailored to general needs and preferences of different populations with lifestyle-related chronic diseases, but the online modality with combined screening and treatment also allows for various ways of tailoring on the individual patient’s level. At the beginning of the intervention, screening for psychosocial and lifestyle-related difficulties enables a selection of patients that are most likely to benefit from the iCBT treatment [19]. Furthermore, visually represented feedback of screening results in personal profile charts gives both patients and their health professionals insights into individual health status and lifestyle, and into specific areas that may need attention [19]. As such, a screening tool with visualized feedback may form an easily implementable tool at a reasonable cost [50], which in itself may already be helpful as a first step in behavior change and as a guide for referral to treatment that suits a patient’s needs [51,52]. A screening tool should be as brief as possible for feasibility reasons. Although the questionnaire set that was composed in this research setting is rather extensive, it should be emphasized that it can be shortened to tailor the tool to clinical practice. Health professionals and patients can decide which instruments are most useful in specific patient populations. For example, if the PPPQ proves to be a valid and reliable instrument, it can be employed as a very brief tool with minimal burden for patients and health professionals, to detect and discuss an individual patient’s functioning and priorities for improvement in a broad range of areas. Subsequently, within our iCBT treatment, individual tailoring is promoted when the patient and therapist collaborate in setting personally relevant treatment goals and selecting the treatment modules and exercises matching those goals [19]. Additionally, contact frequency, modality, and treatment duration can be adapted to optimize attainment of treatment goals. Reviews of online psychological and self-management intervention studies among patients with chronic somatic diseases showed that guided eHealth interventions, in which therapist guidance aids in tailoring the intervention to an individual patient’s needs, are most effective and best adhered to compared to self-help programs [53,54]. In sum, the combination of screening and treatment, provided in an online modality, may form a valuable opportunity to enhance individually tailored and patient-centered care.

In addition to its opportunities for individual tailoring, another main advantage of eHealth interventions is the improved accessibility of self-management support for most patients, including under-served groups [55]. Evidence supports the effectiveness of eHealth interventions in improving health, self-management, and psychosocial outcomes of under-served populations [56]. At the same time, some vulnerable populations may be disadvantaged by eHealth: Patients do need access to digital devices as well as general skills on a computer and Internet use [45], and it has been found that, for instance, people who were unemployed or with low education benefited less from web-based interventions [55]. To optimize eHealth interventions’ effectiveness and acceptability for individuals in under-served groups, it is recommended to incorporate specific tailoring strategies (e.g., to language, culture, and literacy) and technologies (e.g., simple features or no requirement for Internet access), and to include these populations in each stage of intervention development [56]. The latter is a limitation of the current study, as we did not pay special attention to sufficient involvement of members of under-served groups in the co-creation stages of the eHealth care pathway development. Therefore, our web-based care pathway may not be sufficiently accessible for people with limited eHealth literacy or who do not use electronic devices. Yet, we did develop alternative ways of support for people with limited eHealth literacy, such as paper-and-pencil versions of the screening questionnaires and the profile charts, as well as the possibility to add telephone or face-to-face sessions to the treatment. Regardless, involving more participants than we involved in this study is crucial in later stages of evaluation and continued development, including more diverse and under-served populations.

## 5. Conclusions

This paper outlines the evidence-based and systematic development of an eHealth care pathway for patients with lifestyle-related chronic diseases, to identify and treat psychosocial and lifestyle-related difficulties. The study describes the process of using the BCW framework combined with co-creation to design a screening tool and lifestyle self-management modules, tailored to the target population and to individual patient needs. Prior to implementing this eHealth care pathway in hospital care, studies are needed to evaluate its cost-effectiveness and effectiveness on psychosocial, lifestyle, and health-related outcomes, in populations with different lifestyle-related chronic diseases. Prospective assessment between groups would be useful, including a long-term follow-up assessment [27,29]. To this end, our research team is currently conducting randomized controlled trials among populations with chronic kidney disease and end-stage kidney disease (i.e., E-GOAL and E-HELD studies). Afterwards, to achieve successful implementation in regular healthcare, adaptations may be needed to integrate the eHealth care pathway within a specific medical organization or department.

To conclude, the development stages provided in this paper can help to use and refine existing knowledge and tools alongside newly designed intervention components, and merge this into a complex intervention. This systematic process can be applied to guide future intervention development and forms a fundament for further steps of an intervention’s evaluation, continued development, and implementation.

## Figures and Tables

**Figure 1 ijerph-18-03292-f001:**
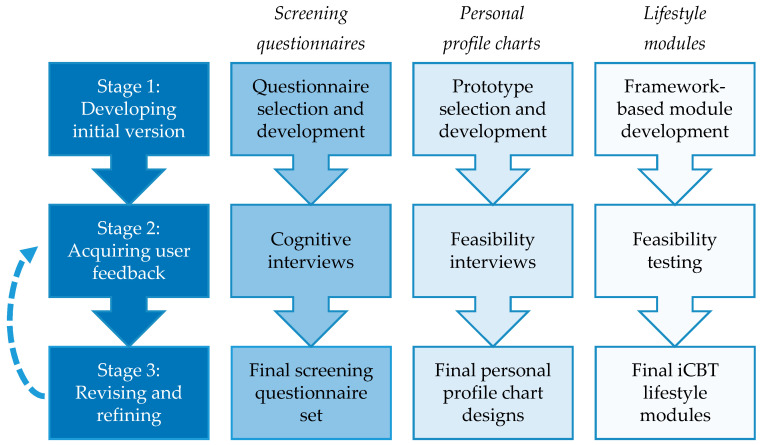
Stages of intervention development per intervention component. iCBT = Internet- delivered cognitive-behavioral therapy.

**Figure 2 ijerph-18-03292-f002:**
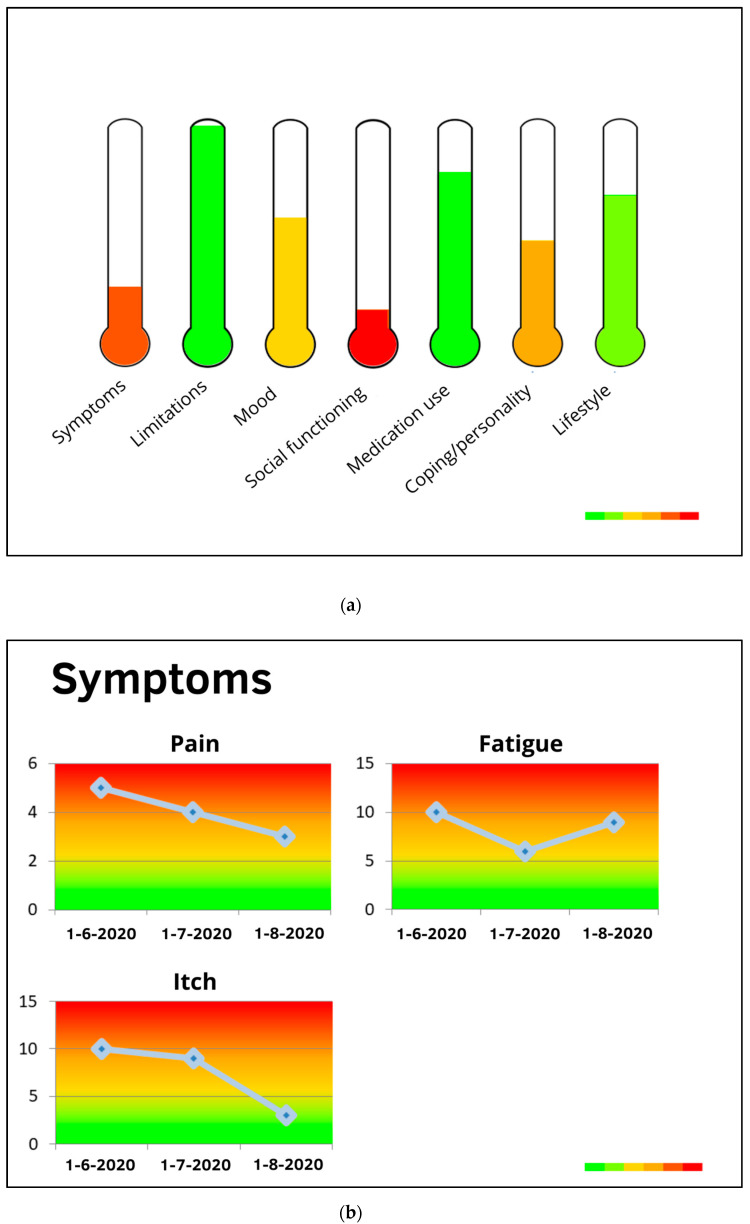
(**a**) Profile chart and (**b**) monitor chart preferred in the first user feedback round.

**Figure 3 ijerph-18-03292-f003:**
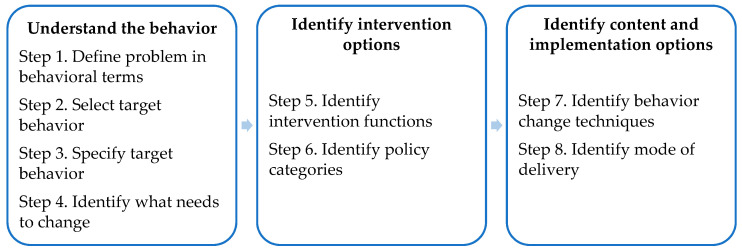
Eight steps of behavior change intervention design. Reproduced with permission from S. Michie, L. Atkins, and R. West, The behaviour change wheel: a guide to designing interventions; UK: Silverback Publishing, 2014.

**Figure 4 ijerph-18-03292-f004:**
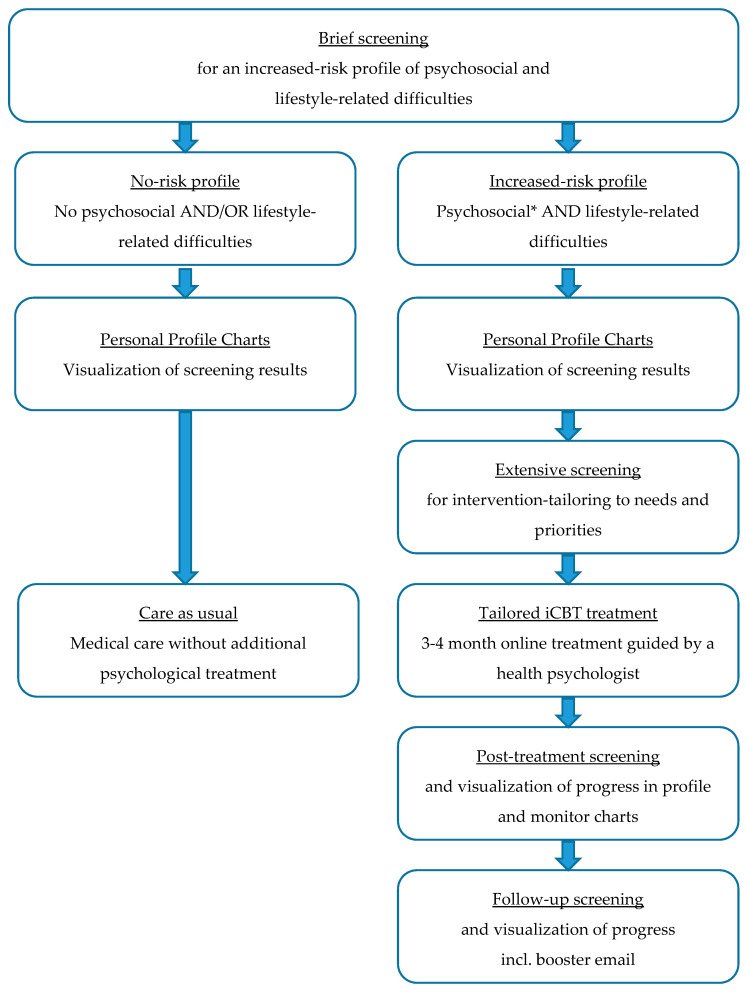
Tailored eHealth care pathway for patients with lifestyle-related chronic diseases. * Patients with severe psychological distress scores are advised to contact their GP for further evaluation and referral to specialized face-to-face mental healthcare.

**Figure 5 ijerph-18-03292-f005:**
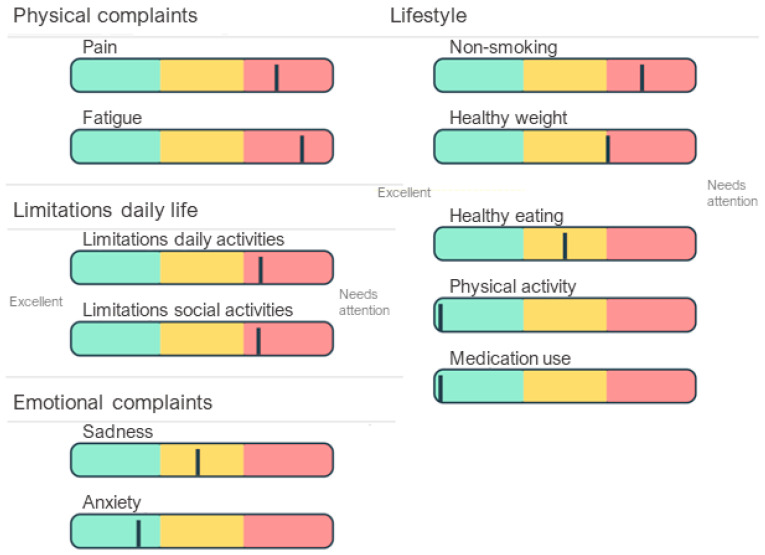
An example of personal profile charts. This patient shows an increased-risk profile with moderate depressive symptoms (which may be influenced by severe physical complaints and limitations in daily life), heavy smoking, obesity, and moderate adherence to dietary prescriptions.

**Figure 6 ijerph-18-03292-f006:**
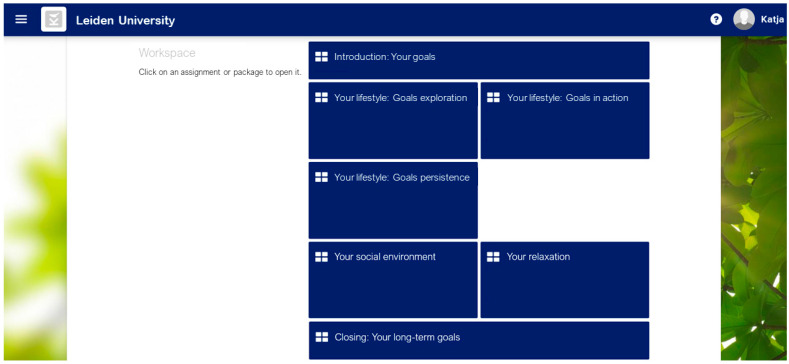
An example of modules in eHealth application “E-coach”.

**Figure 7 ijerph-18-03292-f007:**
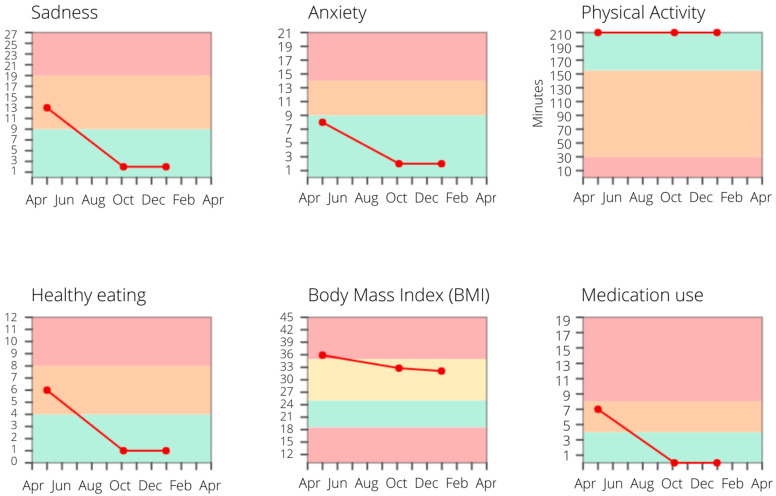
An example of monitor charts. Post-treatment, this patient shows major improvements in depressive and anxiety symptoms, as well as in dietary and medication adherence, which are maintained at follow-up.

**Table 1 ijerph-18-03292-t001:** Specification of the target behavior (Behavior Change Wheel steps 1 to 3). Table template adapted from S. Michie, L. Atkins, and R. West, The behaviour change wheel: a guide to designing interventions; UK: Silverback Publishing, 2014.

Key Behavioral Problem	Unhealthy Lifestyle Behaviors (Leading to Poor Health Outcomes)
What (target behavior)	Physical activity: moderate-to-vigorous intensity ≥150 min per week in multiple sessions Healthy weight: BMI 18.5 to 24.9 Healthy diet: Adherence to dietary prescriptions (e.g., low sodium) Smoking: No tobacco smoking Medication: Adherence to medication prescriptions
Who (target group)	Individuals with lifestyle-related chronic diseases
When/where/how often	Regularly, i.e., on a weekly to daily basis, embedded in daily schedule
With whom	With support from health professionals and social environment

**Table 2 ijerph-18-03292-t002:** Matrix of links between COM-B system, TDF domains, selected intervention functions, and selected BCTs in the lifestyle modules (BCW steps 4, 5, and 7). Matrix template adapted from S. Michie, L. Atkins, and R. West, The behaviour change wheel: a guide to designing interventions; UK: Silverback Publishing, 2014.

COM-B Component (Step 4)	Main TDF Barriers and Facilitators (Step 4)	Selected Intervention Functions (Step 5)	Selected BCTs (Step 7)	Description of BCTs within the Intervention	Lifestyle iCBT Module (Based on Stages of Behavior Change)
Capability	Knowledge (How to engage in a healthy lifestyle)	Education, Persuasion	Instruction on how to perform the behavior; Information about antecedents; Information about health consequences; Feedback on behavior	Guidelines on what, how, and why to engage in healthy lifestyle behaviors; Instruction to keep a record of (unhealthy) behaviors and of events, emotions, and cognitions occurring prior to it; Information about advantages of healthy behaviors; Evaluative feedback on monitored behavior.	1: Goals Exploration (contemplation and decision)
Opportunity	Social influences (Support by professionals and social environment)	Enablement	Social support (unspecified); social support (practical); social support (emotional)	Exercise to discuss personal strengths with important others and how to implement them in behavior change; Exercise to think about ways in which social support is received and about emotional and practical support the person would (not) like to receive; Prompt to ask for support.	3: Goals Persistence (evaluation and maintenance)
	Environmental context and resources (Disease symptoms and material support tools)	Enablement	Restructuring the physical environment; Avoidance/changing exposure to cues for the behavior	Advice and prompt to think about how to avoid exposure to environmental cues for unhealthy behavior and to make adaptations to the environment that facilitate the wanted behavior.	2: Goals in Action (planning and action)
Motivation	Role and identity (Personality characteristics)	Persuasion	Valued self-identity	Exercise to list personal strengths.	3: Goals Persistence (evaluation and maintenance)
	Beliefs about capabilities (Self-efficacy, locus of control)	Persuasion	Focus on past success	Exercise to list previous successes in behavior change.	3: Goals Persistence (evaluation and maintenance)
	Optimism (Acceptance, focusing on possibilities vs. limitations)	Persuasion	Problem solving	Exercise to identify barriers for behavior change and explore ways to overcome them.	1: Goals Exploration (contemplation and decision)
	Emotion (Depression, stress, anxiety)	Education, Persuasion, Enablement	Education, Persuasion: Information on emotional consequences; Self-assessment of affective consequences Enablement: Reduce negative emotions	Information about emotional advantages of healthy lifestyle behaviors; Instruction to keep a record of feelings after performing unhealthy vs. healthy behaviors; Exercise to identify positive self-talk and images to promote positive emotions that facilitate maintenance of the wanted behavior; Exercise to identify ways to reduce negative and stressful emotions.	1: Goals Exploration (contemplation and decision) 3: Goals Persistence (evaluation and maintenance)
	Reinforcement (Noticeable effects of healthy behavior, rewards)	Incentivization	Self-reward, material reward	Prompt to use a personally relevant reward if there has been progress in the wanted behavior.	2: Goals in Action (planning and action)
	Intentions (Intrinsic motivation, joy, higher-order purposes)	Incentivization, Enablement	Pros and cons; Commitment	Exercise to identify and compare reasons for wanting and not wanting to change behavior; Exercise to link the wanted behavior to personally relevant higher-order values; Instruction to write down a decision statement indicating commitment to change behavior.	1: Goals Exploration (contemplation and decision)
	Goals (Concrete and feasible goals)	Enablement	Goal setting (outcome); Goal setting (behavior); Review of outcome goals; Review of behavior goals; Action planning	Exercise to set weekly goals; Instruction to create a daily action (implementation intentions); Prompt to reflect on behavior and correspondence with goals and action plans, leading to re-setting or adapting.	2: Goals in Action (planning and action) 3: Goals Persistence (evaluation and maintenance)
	Beliefs about Consequences (Beliefs about and experiences with consequences of behavior)	Enablement	Pros and cons	Exercise to identify and compare reasons for wanting and not wanting to change behavior.	1: Goals Exploration (contemplation and decision)

COM-B = Capability, Opportunity, Motivation—Behavior; TDF = Theoretical Domains Framework; BCT = Behavior Change Technique.

## Data Availability

The data presented in this study are available on request from the corresponding author. The data are not publicly available due to them containing information that could compromise research participants’ privacy/consent.

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
