# Peer review of "Detecting and Treating Psychosocial and Lifestyle-Related Difficulties in Chronic Disease: Development and Treatment Protocol of the E-GOAL eHealth Care Pathway"

_ijerph, 2021, doi:10.3390/ijerph18063292_

Round 1
Reviewer 1 Report
The authors presented a well-written manuscript on topic of an electronically administered care path for various populations. Creditable effort has already been made but the manuscript still needs fine-tuning in terms of description and generalizability.
Major comments:
- The screening includes at least seven different assessments with probably over 100 different items. It has to be questioned whether the effort that is related to such a “screening” is feasible und suitable for a screening in a real-world setting. Under results, authors speak of “brief screening questionnaires”. Is this a different kind of setting because the setting being described under methods is definitely not brief? How long will it take to complete the screening on average? This may become part of the limitations.
- I have questions about the integration of the screening instruments (Supplement 1). First, the logic behind “PHQ-9 ≥ 20 and/or GAD-7 ≥ 15” is ambiguous, is it “and” or “or”? As described elsewhere, it is presumably “or”. Second, there was no information available how the SF-36 and the SFQ-5 were evaluated nor integrated.
- Regarding the “Personalized Priority and Progress Questionnaire (PPPQ) to measure patients’ personal priorities for improvement as well as actual subjective improvements over time in different areas of functioning (7 items) and lifestyle behaviors (5 items)”, how reliable and valid is it? Since it is mentioned, re-test-reliability and the ability to detect changes over time (sensitivity for changes) is of higher interest to me. It is reported that there will be another manuscript on that topic but it is hard to make reasonable conclusions based on such an instrument prior to its publication and, therefore, this is only of limited use for the current manuscript.
- It is not clear, how the information of questionnaires for intervention-tailoring were used or combined.
- Seriously, there are two pages only on the selection (and development) of the charts. This is excessively much given the fact that the finally selected charts were the only serious contenders among the prototypes.
- Results did not include actual numbers from patients using the module. How many patients have used it so far? Are there basic demographic characteristics available? To how many patients does this system apply to? CBTs are still resource-dependent components.
- Authors claim that “a strength of the eHealth care path is its flexibility for usage in patient populations with different lifestyle-related chronic diseases”. The manuscript does not provide evidence that substantiates this conclusion. The study demonstrates feasibility after an intensive development process. If I am mistaken here and the authors wanted to speak about “an eHealth care path” in general, this should be made clear.
- As the title indicates (E-GOAL), the main focus was placed on a care path for kidney diseases. I was wondering how generalizable the developed tool truly is/will be. In other parts of the manuscript, it is described as a generic tool. There is a little tension throughout the manuscript between the desire to be applicable to many domains and the origin in nephrology. The manuscript would benefit from a more rigorous decision for one of these two views.
- How is (or will be) the iCBT reimbursed?
Minor comments:
- Use „web-based“ instead of „internet-based“. Further, Internet always starts with a capital. letter.
- Add a more precise definition for “eHealth care path” in the introduction.
- Where is Table 1?
Reviewer 2 Report
I would like to congratulate the authors on a wonderfully articulate, clear and, considering the amount of ground covered, concise yet detailed manuscript. From start to finish I felt entirely clear on what your aims, processes and findings were. The introduction convincingly justifies the need for this dual focus on both psychological distress and lifestyle behaviour, the methodology is coherently described and makes good use of tables and figures to bring the content to life, and the discussion is commensurate with the findings and study design. All in all, just a few minor points to consider from me:
Line 25 (abstract) and 565 (discussion): this may well be up for debate but I wonder if it is accurate to describe TDF/BCW as 'theoretical frameworks' as they are not theories of behaviour change per se, but rather process frameworks that guide the development and implementation of interventions. Would 'theory-based-', or 'theory-informed-' frameworks be more appropriate?
Line 85: I find the phrase 'increased risk for long-term adjustment problems...' ambiguous. I guess you mean people who may resist change to the their lifestyle? Perhaps a more explicit phrase could be used.
Line 170: You might consider moving the first line under sub heading Part 1: Questionnaires for Increased Risk Profile ID "In order to limit the burden...two subsequent parts." up to subheading 2.2.1 so it appears before the next sub heading Part 1... I feel this will enhance readability.
Line 176-178: Was there a theoretical or empirical basis for choosing these particular psychosocial difficulties that you include in the screening that you can cite here? It feels like the justification of why these is overlooked in the main text.
Line 199: It would be useful for you to list a few examples of the scales here used in the PPPQ.
Line 232: it is not clear if the 'subsequent interviews with 4 participants' are new patients or a re-interview of those who took part in the first iteration. It would be useful to clarify.
Line 284: I was not sure from the description whether the participants just indicated pluses and minuses (i.e. symbols) on the graphs that were then discussed with the interviewer, or if they wrote specific positive and negative comments on the graphs. Either way this could be clearer.
Line 392: suggest paragraph break at 'Step 5' as very long paragraph otherwise
Line 443: It would be useful to specify what the chosen intervention duration was here.
Figure 7: the legibility of axis headings and scales isn't particularly good (on my reviewed version anyway) and the physical activity 100's do not appear.
Line 637: This is a bit pedantic as I very much value the fact that you have used some rather than no co-creation at all, which will have no doubt improved the tool (and I recommend you caveat any limitation with this), but I wonder if you do need to acknowledge in your limitations section that some stages of feedback involved very few participants (i.e. 2 or 4) and so it would be unrealistic to expect their opinions to generalise or be representative of a wider, more diverse population. I think adding here, or to your concluding remarks that evaluating effectiveness in larger and more diverse samples, including those who are underserved and may be less amenable to using eHealth is a crucial next step for the E-GOAL tool.
Reviewer 3 Report
The work is a presentation of a practical work protocol with people exposed to civilization diseases. The tone of the material is more practical, but I also see a lot of possibilities for collecting scientific data. Therefore, I believe that it should be presented primarily as an e-intervention tool. One of the important limitations, apart from those indicated by the authors, is the limited access of people who do not use electronic tools.
